# Domain Generalization in-the-Wild: Disentangling Classification from Domain-Aware Representations

## Abstract

Evaluating domain generalization (DG) for foundational models like CLIP is challenging, as web-scale pretraining data potentially covers many existing benchmarks. Consequently, current DG evaluation may neither be sufficiently challenging nor adequately test genuinely unseen data scenarios. To better assess the performance of CLIP on DG in-the-wild, a scenario where CLIP encounters challenging unseen data, we consider two approaches: (1) evaluating on 33 diverse datasets with quantified out-of-distribution (OOD) scores after fine-tuning CLIP on ImageNet, and (2) using unlearning to make CLIP 'forget' some domains as an approximation. We observe that CLIP's performance deteriorates significantly on more OOD datasets. To address this, we present CLIP-DCA (**D**isentangling **C**lassification from enhanced domain **A**ware representations). Our approach is motivated by the observation that while standard domain invariance losses aim to make representations domain-invariant, this can be harmful to foundation models by forcing the discarding of domain-aware representations beneficial for generalization. We instead hypothesize that enhancing domain awareness is a prerequisite for effective domain-invariant classification in foundation models. CLIP-DCA identifies and enhances domain awareness within CLIP's encoders using a separate domain head and synthetically generated diverse domain data. Simultaneously, it encourages domain-invariant classification through disentanglement from the domain features. CLIP-DCA shows significant improvements within this challenging evaluation compared to existing methods, particularly on datasets that are more OOD.

## 1 Introduction

Domain generalization (DG) aims to train models that maintain robust performance when encountering out-of-distribution (OOD) data [1]. A key assumption of DG is that the target domains represent novel data distributions for evaluation. However, this assumption is challenged when evaluating pretrained foundation models like CLIP [2] and ALIGN [3]. These models have been trained on comprehensive web-scale datasets, thus have likely been exposed to most existing domains, contributing to its impressive zero-shot capabilities. Consequently, much research has focused on adapting CLIP through parameter-efficient finetuning [4, 5, 6, 7], regularization using the original weights [8, 9, 10, 11], and even transductive methods [12, 13], largely preserving its pretrained knowledge. However, a critical question arises: **how would CLIP perform on genuinely unseen domains?** A recent study [14] found that retraining CLIP from scratch using only natural images significantly degrades performance on OOD benchmarks, resulting in performance similar to models trained only on ImageNet. Our results (Figure 1) align with these findings, suggesting current DG evaluations for CLIP may overestimate its true OOD robustness because standard evaluation settings like leave-one-domain-out and existing cross-dataset evaluations may not be sufficiently challenging (Sec. 4.1, 4.2).

Submitted to 39th Conference on Neural Information Processing Systems (NeurIPS 2025). Do not distribute.

We therefore propose that DG evaluation for foundation models, such as CLIP, should be more challenging, to approximate "domain generalization in-the-wild," where CLIP might encounter diverse and challenging new data in the real-world. We evaluate CLIP on 33 target datasets spanning a diverse range of OODness. To systematically approach evaluation, we quantify a multi-modal OOD score (Sec. 4.2), using ImageNet as both an anchor and a source dataset owing to its inclusion of many classes and concepts. We find that after finetuning on ImageNet, CLIP's DG performance degrades on datasets with higher OOD scores with respect to ImageNet (Figure 1), consistent with the domain contamination findings [14]. In addition, to further simulate truly unseen domains, we use an unlearning technique [15] to make CLIP forget some domains (Sec. 4.3), and find significant performance degradation for existing robust finetuning methods.

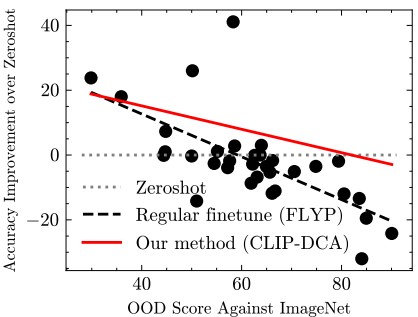

Figure 1: Improvement over zeroshot after finetuning on ImageNet (in %). OOD scores are quantified relative to ImageNet (source dataset), illustrating the challenge of DG in-the-wild.

Our results (Figure 9), alongside findings on domain contamination [14], suggest that for DG in-the-wild, different robust finetuning algorithms are needed for genuinely unseen data. In light of this, we present CLIP-DCA (**D**isentangling **C**lassification from enhanced domain **A**ware representations), an end-to-end finetuning method to improve the robustness of CLIP on truly OOD data. A key idea in DG is that learning domain-invariant features is beneficial for robust generalization [1, 16]. However, naively enforcing domain invariance for a pretrained foundation model could cause catastrophic forgetting of useful features learned from diverse domains during pretraining as the model is forced to make its representations entirely domain-invariant. We hypothesize that to learn effective domain invariance, domain awareness is a prerequisite. This awareness is critical to maintain CLIP's vast knowledge, which includes generalizable features that support capabilities like zero-shot classification. By enhancing domain awareness, CLIP can also selectively disentangle classification from domain-specific aspects, thereby achieving robust generalization without forgetting valuable information.

We combine the idea of domain awareness and domain invariance by encouraging them simultaneously within CLIP-DCA (Figure 2). Specifically, we encourage domain awareness within CLIP's image and text encoders, while promoting domain invariance specifically at the final classification layer through disentanglement. Our premise is that while domain awareness is a requirement to maintain pre-existing knowledge, this awareness can be disentangled for domain-invariant classification and robust generalization. To achieve this, we add a new head to the CLIP image encoder, called the domain head, which is trained to understand domains. The original classification head is then disentangled from the domain head, effectively learning domain awareness within its encoders and achieving domain invariance at the classification stage. Additionally, since many datasets lack distinct domains or textual descriptions, and the definition of 'domain' is often vague in DG in-the-wild, we address this by using diffusion models to create images of artificial domains and MLLMs to generate descriptions for these artificial domains (Sec. 3.2). Our contributions are summarized as follows:

- We demonstrate potential limitations in current DG evaluations of foundation models, supported by our results and a recent study. Existing benchmarks may overestimate true OOD robustness, potentially leading finetuning strategies towards in-distribution improvement rather than OOD.
- We propose more challenging and holistic evaluations for DG in-the-wild. We use an expanded cross-dataset evaluation setting spanning 33 datasets from diverse domains, indexed by multi-modal OOD scores. We also use an unlearned model to further approximate unseen domains.
- We introduce CLIP-DCA, a novel finetuning method that improves OOD robustness by disentangling classification from enhanced domain-aware representations. We find that on more OOD target datasets, CLIP-DCA performs significantly better compared to existing robust finetuning methods, while performance is similar across all methods on less OOD target datasets.

## 2   Related Work

**Domain Generalization.** Learning domain-invariant representations has historically been a central idea in domain generalization [17, 1]. The intuition is that when classifying images from entirely

new distributions, learning abstract features common across source domains should provide better robustness for classification in new domains [17, 18]. Among these, domain-adversarial learning methods have become a relatively standard approach within the DG field due to its conceptual simplicity and effectiveness [1]. For instance, Domain Adversarial Neural Networks (DANN) [16] uses an auxiliary domain classifier trained adversarially against the encoder, encouraging the encoder to produce features indistinguishable across source domains. Given the focus of DANN on the central idea of domain invariance, we focus on DANN and its adaptation to CLIP in our analysis. Notably, despite the prevalence of such DG methods, the direct application for CLIP is not well-established and remains underexplored. Naively enforcing domain invariance on foundation models like CLIP, with large pretrained knowledge, risks catastrophic forgetting.

**Robust Finetuning of CLIP.** The introduction of CLIP marked a significant shift in DG research. The original study [2] demonstrated impressive zero-shot classification performance across diverse benchmarks, including OOD datasets. The authors attributed this capability to CLIP learning representations that are less reliant on spurious correlations specific to downstream target datasets, as CLIP was not trained on these specific datasets during its initial pretraining.

The assumption of the inherent OOD robustness in CLIP motivated numerous methods aimed at finetuning CLIP for downstream tasks while enhancing its perceived robustness. A common approach is parameter-efficient finetuning (PEFT) strategies. An early influential study, CoOp [4], introduced learnable textual prompts, motivated by observations that manually crafted prompt ensembles improved CLIP's zero-shot accuracy. Building on this, CoCoOp [5] made these prompts dynamic by conditioning them on individual image features through a cross-attention mechanism. Similarly, CLIP-Adapter [6] proposed adding lightweight, learnable MLP layers (adapters) to the CLIP encoders, finetuning only these small adapters instead of the entire network. Many more subsequent PEFT methods have also been explored [19, 20, 21, 22, 23, 24, 25, 26, 27].

End-to-end finetuning methods have also been explored, yet many still depend on the original pretrained CLIP weights for regularization or guidance. Wise-FT [8], motivated by observing that standard finetuning often degraded zero-shot OOD performance, ensembles the weights of the finetuned model with the original CLIP weights. CLIP-OOD [11] used a beta-moving average of the weights during finetuning alongside a regularization term to enhance semantic relationships learned during pretraining. MIRO [28] used mutual information regularization between the finetuning model and the frozen pretrained CLIP model to retain pretrained features.

While many other methods show strong performance on OOD benchmarks, this overview highlights representative approaches, their trends, and assumptions in robust CLIP finetuning. Our work, however, *questions whether current evaluation protocols are sufficiently challenging, and suggests the reliance on the pretrained weights may be suboptimal for true OOD generalization*, a concern supported by evidence of domain contamination during pretraining [14]. Consequently, we explore more challenging evaluations and alternative strategies for training CLIP grounded in DG principles.

# 3   CLIP-DCA: Disentangling Classification from Enhanced Domain-Aware Representations

The previous sections have highlighted the complexity of evaluating domain generalization in foundational models like CLIP. In light of this, we propose a novel finetuning approach, CLIP-DCA (**D**isentangling **C**lassification from enhanced domain **A**ware representations), designed to improve robustness by balancing the trade-off between domain invariance and knowledge retention.

## 3.1   Encouraging domain awareness and invariance simultaneously

Our key hypothesis is that domain invariance at the decision-making stage is beneficial for generalizing to unseen domains. At the same time, domain awareness is required for retaining the vast pretrained knowledge of CLIP. We achieve them simultaneously by encouraging domain awareness in the encoders, while enforcing domain invariance only in the classifier of CLIP through disentanglement. The intuition is that **if a model understands what constitutes as domain-specific features, then it can learn to disregard it appropriately during classification on unseen domains**.

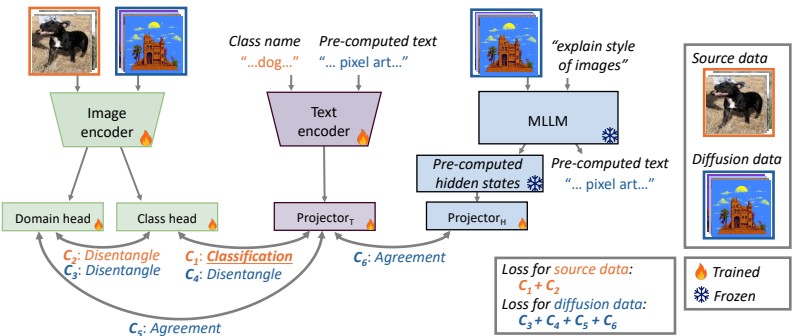

Figure 2: CLIP-DCA applies different sets of losses to source data images and diffusion images. For source images, accurate classification is encouraged through the classification loss between class head and text encoder ($C_1$). Invariance is encouraged through the disentanglement between domain and class heads ($C_2$). With diffusion images, domain invariance is encouraged through the disentanglement between the domain and class heads ($C_3$), and disentanglement between class head and text encoder ($C_4$). Domain awareness is encouraged through the agreement between the domain head and the text encoder ($C_5$), and the agreement between the text encoder and the MLLM hidden states ($C_6$). During inference, only the class head and text projector are used for classification.

Enforcing domain invariance in the encoder through conventional domain adversarial learning, for instance, can be harmful. Our experiments show that applying invariance directly leads to worse performance compared to standard finetuning (Figure 8). Forcing the entire model to become domain-invariant can lead to the forgetting of valuable, fine-grained features learned during the pretraining on a large dataset. Conversely, existing CLIP robust finetuning methods discourage divergence from the original pretrained model, and rely on the assumption that CLIP is inherently robust to OOD data. This assumption is challenged by our results (Figure 9) and evidence for domain contamination [14].

Instead, we focus on enforcing domain invariance only at the final classification layer, while simultaneously encouraging the image encoder to become domain-aware. Our intuition is that a comprehensive understanding of various domains enables the model to more effectively disregard domain-specific influences during inference. The diverse set of generated diffusion images and their descriptions (detailed in Section 3.2) provides the necessary signals for enhancing this domain awareness.

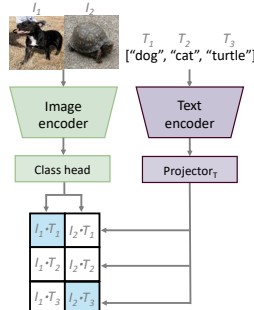

Figure 3: Standard CLIP inference pipeline using a dot product between image and text embeddings for classification.

To implement this, we introduce an architectural addition to the CLIP image encoder. We add an additional linear projection head, termed the image domain head ($I_D$), which has the same dimensionality as the original image projection head, referred to as the image class head ($I_C$), as shown in Figure 2. We do not add a corresponding domain head to the text encoder for two reasons. First, in most downstream classification datasets, only class names are available as text inputs, without domain descriptions. Second, textual information inherently allows for easier separation of domain and class attributes. For instance, a prompt like "a sketch of a dog" clearly distinguishes class ("dog") from domain ("sketch"). Note that for inference, the standard pipeline is used as shown in Figure 3. The domain head and other losses are not used.

During training, we use two distinct loss functions for the two types of data we use - the source dataset and generated diffusion images. We use $\ell_a$ to refer to agreement loss (the standard CLIP contrastive loss [2] or finetuning [29]). We use $\ell_d$ to refer to disentanglement, which we define as the squared sum of the diagonal of the cross-correlation matrix — $\ell_d = \sum_i ((XY^\top)_{ii})^2$, where $X$ and $Y$ are normalized matrices of size [batch, feature]. We also use $P_T$ to refer to the projected embeddings of the text encoder, and $P_H$ as the projected hidden states of the MLLM.

We simultaneously encourage accurate classification, domain awareness in both text and image encoders, and domain invariance at the classification stage with the following loss terms:

1. **For the source dataset images (e.g., ImageNet, with only class labels):**

- A *classification loss* (i.e., the standard CLIP contrastive loss [29]) between the output of the image class head and the text embedding of the class name, $C_1 := \ell_a(I_C, P_T)$.
- A *disentanglement loss* between the class and domain heads, $C_2 := \ell_d(I_C, I_D)$.
- For source dataset images, we minimize the loss function $\mathcal{L}_{source} = C_1 + C_2$.

2. **For the diffusion images and their MLLM-generated style descriptions:**
   - A *disentanglement loss* between the class head and domain head, $C_3 := \ell_d(I_C, I_D)$.
   - A *disentanglement loss* between the text embedding of style descriptions and the image class head to further encourage the class head to learn domain invariance, $C_4 := \ell_d(P_T, I_C)$.
   - An *agreement loss* between the output of the image domain head and the text embedding of the style description, enhancing domain head's domain awareness, $C_5 := \ell_a(P_T, I_D)$.
   - An *agreement loss* between the text embedding and the corresponding projected MLLM hidden state, enhancing the text encoder's domain awareness, $C_6 := \ell_a(P_T, P_H)$.
   - For diffusion images, we minimize the loss function $\mathcal{L}_{diffusion} = C_3 + C_4 + C_5 + C_6$.

## 3.2 Generating diverse domains

Traditional DG benchmarks provide multi-domain datasets, enabling the learning of domain invariance. However, our evaluation setup, which involves finetuning on a single source dataset like ImageNet, lacks explicit multiple source domains, especially as the boundary for different domains becomes more vague for DG in-the-wild. Additionally, we hypothesize that to understand what constitutes as domain-specific features, a diverse number of domains are required.

To address this, we construct a small dataset with a diverse number of domains. As illustrated in Figure 4, we prompt a Multimodal Large Language Model (MLLM), specifically LLaVA [30], to generate ideas of 512 distinct styles for images (e.g. "pixel art"). A text-to-image diffusion model (Stable Diffusion 3 [31]) then generates images from these stylistic prompts. We intentionally omit any class labels during image generation to ensure the styles are not biased towards specific classes. We generate 8 images per style, creating a dataset of 4096 images. Finally, the same MLLM generates textual domain descriptions (captions) for each style. We also store the hidden state representations from the MLLM that were used to generate these style descriptions, as these will be used to encourage domain awareness in the text encoder.

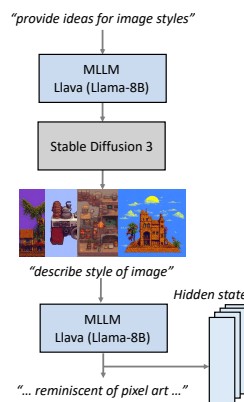

Figure 4: Pipeline for generating synthetic domain images and descriptions.

# 4 Experimental Setup

## 4.1 Evaluating DG in-the-wild performance

While standard domain generalization benchmarks such as DigitsDG [32], PACS [33], Office-Home [34], Terra Incognita [35], FMOW-Wilds and Camelyon-Wilds [36], and ImageNet variants [37, 38, 39, 40, 41] are widely used, we observe that the different domains within a single benchmark dataset often exhibit greater similarity to each other than to conceptually similar domains across different datasets. To quantify this, we use Spectral-normalized Neural Gaussian Process (SNGP) [42] to compute the pairwise OOD scores between domains across these benchmarks. We then visualized the pairwise OOD scores with PCA. This analysis reveals significant clustering within individual benchmarks as shown in Figure 5. The clustering within benchmarks, combined with the impressively high zeroshot accuracy, and the success of transductive methods on certain DG datasets [12, 13], provides strong evidence that the current DG evaluation is not very difficult for CLIP, possibly due to domain contamination.

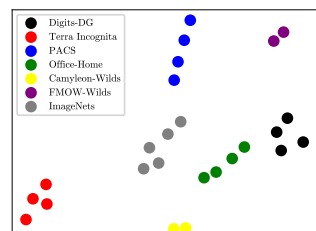

Figure 5: PCA visualization of domains from different domain generalization datasets

Consequently, we evaluate using a more challenging cross-dataset setup aiming to simulate DG in-the-wild. We finetune the model on ImageNet-1K [43], and evaluate its generalization capabilities across a wide range of 33 target datasets, including the standard DG benchmarks listed above. A

cross-dataset evaluation is significantly more challenging compared to traditional DG setups, as it involves larger visual distribution shifts and also shifts in class labels. This evaluation also aligns with the methodologies of prior studies investigating robust CLIP finetuning [4, 5, 6, 11], while adding a broader coverage of domains. We use the CLIP ViT-B/32 model for all experiments. Other training details are included in the appendix.

## 4.2 Measuring OODness of the target datasets

Given that our DG in-the-wild evaluation includes many target datasets with varying degrees of OODness compared to ImageNet, establishing a quantitative OOD metric is beneficial for a more holistic assessment of OOD robustness. A unique consideration for CLIP is its dual-encoder architecture. To provide a comprehensive score, we utilize OOD measures for both the image and text modalities. For the image encoder, we use SNGP [42] calibrated on the ImageNet validation data to compute an OOD score for all 33 target datasets. In addition, we use a text-based OOD measure [44] to measure OODness of class labels. This involves calculating classification probabilities on a combined label set of target dataset class names and ImageNet class names using the target domain image embeddings. The text OOD score is the summed probability assigned to the target-specific class names.

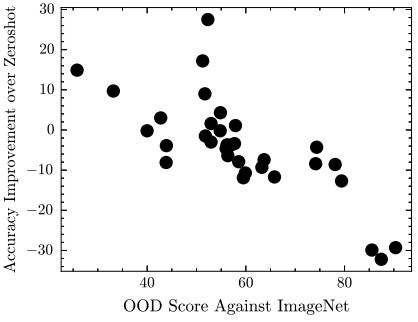

Figure 6: OOD score of 33 target datasets against ImageNet and classification accuracy improvement over zeroshot

We verify that our OOD score shows a strong negative correlation (r=-0.756, p<0.001) with performance on target datasets after finetuning, as shown in Figure 6. Notably, we find that averaging the image and text OOD scores is important for accurately predicting post-finetuning accuracy. Relying solely on the image OOD score (r=-0.099) or the text OOD score (r=-0.608) yields weaker correlations, providing evidence that OOD scores in both modalities are necessary for a comprehensive understanding of OOD challenges in the context of CLIP.

## 4.3 Approximating unseen data through unlearning

Ideally, evaluating true DG performance would involve using a CLIP model trained without specific target-like domains. Unfortunately, retraining CLIP from scratch while omitting certain data is computationally expensive due to the scale of original training data (millions to billions of images). Public weights for selectively trained models are unavailable.

Given these constraints, we explore an approximate approach inspired by the concept of unlearning to mitigate potential domain contamination. Specifically, we adapt the adversarial learning-based unlearning method [15] for domain forgetting. We finetune CLIP [29] using a dual objective. First, to retain general knowledge, we train on a 595,000-image subset of the CC3M dataset [45], referred to as GCC, previously used in LLaVA pretraining [30], serving as a manageable proxy for CLIP's original training data. Second, to approximate a scenario where domains similar to DomainNet are removed, we apply domain adversarial training [16] on the DomainNet dataset, which we exclude from our target datasets. We attach a binary classifier to the penultimate layer of the image encoder. During training batches, this classifier is fed representations of random noise (assigned label 0) and images from DomainNet (assigned label 1). The gradient reversal layer [16] forces the image encoder to learn representations that confuse this classifier, making embeddings of DomainNet images and random

Table 1: Performance of ZS (zero-shot), FT (Regular finetuning on GCC), and Unlearn (Regular finetuning on GCC + unlearning on DomainNet).

| Metric/Data | ZS | FT | Unlearn |
| --- | --- | --- | --- |
| ***Imagenet*** | | | |
| IN 1 | 54.2 | 52.0 | 48.8 |
| IN 2 | 48.4 | 45.5 | 41.8 |
| IN Sketch | 32.3 | 31.5 | 30.7 |
| IN A | 26.2 | 19.0 | 18.2 |
| IN R | 59.7 | 56.8 | 52.7 |
| | | | |
| ***DomainNet*** | | | |
| Clipart | 64.3 | 67.0 | 53.0 |
| Infograph | 41.6 | 41.0 | 34.0 |
| Painting | 54.4 | 53.9 | 47.0 |
| Real | 80.5 | 80.7 | 73.3 |
| Sketch | 57.9 | 57.2 | 45.5 |
| Quickdraw | 12.1 | 8.2 | 0.3 |
| Avg. on 33 | 51.1 | 49.7 | 45.5 |

noise indistinguishable, thereby encouraging the model to unlearn domain-specific features from DomainNet. The unlearning occurs concurrently with standard training on the GCC dataset to preserve CLIP's core capabilities.

We intentionally avoid unlearning the target datasets used in our evaluation. This ensures a fairer comparison against existing robust CLIP finetuning methods, many of which discourage heavily diverging from the original pretrained weights. Directly unlearning the target datasets would give the baselines an unfair disadvantage. Instead, unlearning DomainNet serves as a proxy for reducing domain contamination effects. We find that performance on DomainNet and some other datasets drops, as shown in Table 1, while retaining much of the performance on many other datasets.

## 5 Results and Discussion

### 5.1 Finetuning original pretrained CLIP

We first evaluate CLIP-DCA in the context of our domain generalization in-the-wild setup, using the original pretrained CLIP weights as the starting point. As shown in Figure 7, CLIP-DCA consistently improves performance over standard finetuning across target datasets. Importantly, the best-fit line for CLIP-DCA shows a flatter slope, indicating that it is more robust to more severe OOD data compared to regular finetuning. This observation aligns with our hypothesis that encouraging domain invariance at the decision-making layer, while simultaneously encouraging domain awareness within the encoders, is crucial for robust classification on unseen distributions.

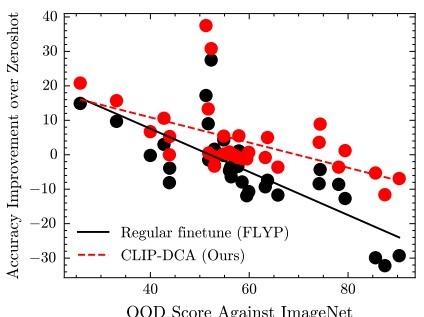

Figure 7: Performance comparison of CLIP-DCA against regular finetuning. Best-fit lines, determined by linear regression, illustrate performance trends.

Figure 8 provides a broader comparison against additional baselines. We observe that conventional domain adversarial learning (DANN [16]), is harmful for CLIP, showing inferior performance compared to regular finetuning. This shows the potential disadvantage of enforcing domain invariance across the entire image encoder, which can lead to excessive forgetting of features learned during pretraining. This suggests the importance of approaches such as our proposed learning of targeted invariance through disentanglement.

Interestingly, on the most extremely OOD datasets, parameter-efficient finetuning (PEFT) techniques like CoOp [4] and CLIP-Adapter [6] perform best. PEFT methods minimally change a small subset of the original CLIP weights. Consequently, their performance shows much lower variance across the datasets, with improvements (around 1-2%). It is important to note that on extreme OOD datasets, all end-to-end finetuning methods exhibit lower performance than the zero-shot CLIP baseline. While CLIP-DCA mitigates this performance drop compared to standard finetuning, it does not entirely overcome it.

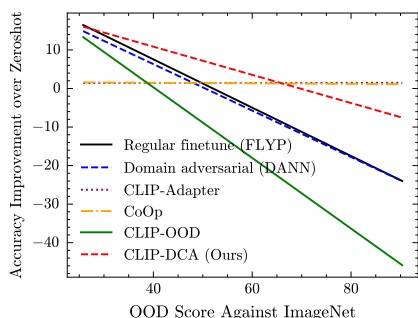

Figure 8: Comparison against more baselines

This strong zero-shot performance has often been attributed to CLIP's inherent OOD generalization capability. However, the study by [14] challenges this assumption and shows that this generalization could be attributed to domain contamination. They show that when CLIP is retrained solely on natural images, its OOD performance drops to similar levels as models trained exclusively on ImageNet. This drop could offer a plausible explanation for observations like those motivating Wise-FT [8], where standard finetuning was found to degrade OOD performance.

### 5.2 Finetuning after unlearning

To further investigate the impact of potential domain contamination and to establish a more rigorously "unseen" evaluation, we applied the unlearning procedure detailed in Section 4.3 to the pretrained

Table 2: Accuracy on ImageNet variants

| Method | V1 | V2 | Sketch | A | R |
|---|---|---|---|---|---|
| Zeroshot (unlearned) | 48.8 | 41.8 | 30.7 | 18.2 | 52.7 |
| Regular Finetune | 69.8 | 58.4 | 34.7 | 15.0 | 52.6 |
| DANN | 70.0 | 58.2 | 33.2 | 16.5 | 52.0 |
| CLIP Adapter | 52.9 | 45.7 | 28.4 | 15.0 | 51.6 |
| CoOp | 53.3 | 46.2 | 29.1 | 16.1 | 52.8 |
| Wise-FT | 72.9 | 61.3 | 40.0 | 9.4 | 43.0 |
| MIRO | 74.1 | 62.7 | 35.7 | 7.3 | 33.2 |
| CLIP-OOD | 69.0 | 58.2 | 35.3 | 15.0 | 45.8 |
| **CLIP-DCA (Ours)** | **75.1** | **63.9** | **42.2** | **22.9** | **62.2** |

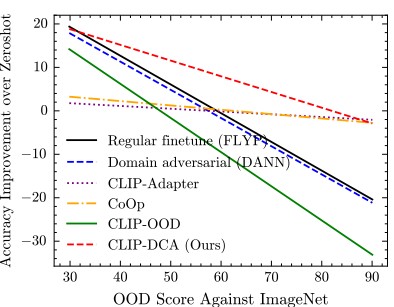

Figure 9: Comparison against baselines after unlearning

CLIP model. We then finetuned this "unlearned" model on ImageNet-1K and evaluated its performance. Table 2 shows the accuracies on the ImageNet variant datasets. For this analysis, we also include several end-to-end robust finetuning methods that add a linear classifier to CLIP. Due to their architecture, these specific baselines are evaluated only on the ImageNet variants as they cannot be adapted to datasets with different class labels.

Our results show that robust end-to-end finetuning methods remain effective for datasets that are less OOD even after unlearning. For instance, MIRO [28] and Wise-FT [8] outperform regular finetuning on ImageNet-V1, ImageNet-V2, and ImageNet-Sketch. However, consistent with the trends seen with the non-unlearned model, performance significantly drops on datasets with larger OOD scores, such as ImageNet-A and ImageNet-R. Similarly, PEFT methods show slight improvements over the unlearned zero-shot baseline on ImageNet-V1, V2, and Sketch, but their performance drops on ImageNet-A and R.

Figure 9 shows that the performance of all methods, even PEFT methods, further drops as OODness increases across target datasets when finetuning the unlearned model. If the unlearning process successfully reduced the knowledge of target-like domains, existing robust finetuning methods, which rely on the pretrained weights, would struggle on genuinely OOD data. These results suggest that our unlearning approach was effective in simulating a less contaminated starting point.

With the unlearned model, CLIP-DCA shows high performance. For datasets with moderate OOD scores relative to ImageNet, CLIP-DCA achieves larger performance improvements compared to other methods. More importantly, on the extremely OOD datasets, the performance of our method remains close to the zero-shot model, without significant performance drops. This suggests that our mechanism of encouraging domain awareness while selectively enforcing invariance at the decision layer is particularly beneficial when starting from a model with reduced prior exposure to target-like domains.

Table 3: Ablations on GCC inclusion. Accuracy on ImageNet variants (V1, V2, Sketch, A, R) and Avg. accuracy on 33 datasets.

| Setting | V1 | V2 | Sketch | A | R | Avg. |
|---|---|---|---|---|---|---|
| Zeroshot | 46.0 | 40.4 | 27.4 | 15.1 | 51.6 | 45.5 |
| *ImageNet only* | | | | | | |
| FLYP | 69.8 | 58.4 | 34.7 | 15.0 | 52.6 | 43.6 |
| DANN | 70.0 | 58.2 | 33.2 | 16.5 | 52.0 | 42.5 |
| **CLIP-DCA** | **75.3** | **64.1** | **40.3** | **22.3** | **60.3** | **48.6** |
| *ImageNet+GCC* | | | | | | |
| FLYP | 70.6 | 59.7 | 38.5 | 17.6 | 57.5 | 49.0 |
| DANN | 70.5 | 59.4 | 38.6 | 17.4 | 57.2 | 47.5 |
| **CLIP-DCA** | **75.1** | **63.9** | **42.2** | **22.9** | **62.2** | **52.1** |

### 5.3 Ablations

**Including GCC data.** When finetuning CLIP-DCA, we also use the GCC dataset – the dataset with 595,000 image-caption pairs used to prevent CLIP from collapsing during the unlearning procedure (Sec. 4.3). While the dataset is smaller than ImageNet-1K, it serves as a manageable proxy for the data CLIP was originally pretrained on. The image-caption pairs provide valuable supervision particularly for training the text encoder and possibly preventing catastrophic forgetting during finetuning on a classification datasets like ImageNet.

We study the contribution of the GCC data as shown in Table 3. A key observation is that the inclusion of GCC provides a notable benefit even for standard finetuning (FLYP) [29]. This shows the general benefit of incorporating diverse, captioned data during finetuning. Given these benefits, an alternative or complementary approach could involve using MLLMs to generate rich textual descriptions for

classes or images within the primary source dataset, similar to strategies explored in [46, 47], which use an LLM to describe class names. Despite the general improvements, our method consistently shows higher performance even when the GCC dataset was not included.

**Different components of CLIP-DCA.** We study the effect of the different components of CLIP-DCA, as shown in Table 4. We isolate the use of domain descriptions from diffusion images to train the image domain head, the disentanglement loss between the class and domain heads to encourage invariance at the classifier, and the use of MLLM hidden states to encourage domain awareness in the text encoder. Simply introducing domain descriptions to make the image encoder aware of styles, without enforcing disentanglement at the classifier, shows only a marginal improvement over the FLYP baseline, suggesting that *domain awareness alone is insufficient without*

Table 4: Ablation of CLIP-DCA components: Domain descriptions (Domain), Disentanglement (Disent.), MLLM Hidden States (MLLM HS), and Avg. accuracy on 33 datasets.

| Method / Config. | Domain | Disent. | MLLM HS | Avg. |
|---|---|---|---|---|
| MLLM (LLaVA) | - | - | - | 24.2 |
| FLYP | X | X | X | 49.0 |
| Ours | O | X | X | 49.1 |
| | O | O | X | 50.8 |
| | O | X | O | 49.0 |
| **Our full** | **O** | **O** | **O** | **52.1** |

*a mechanism to disentangle classification from it*, as CLIP may otherwise struggle to disregard domain-specific features irrelevant to classification. When we incorporate the disentanglement loss to encourage domain invariance at the decision-making layer, even without explicit domain awareness in the text encoder, performance slightly improves. This is further evidence for our core hypothesis that enabling the model to disregard domain-specific features during classification is important. Attempting to make both encoders domain-aware without the disentanglement loss results in no improvement over the baseline, indicating that awareness without a mechanism for invariance can be ineffective for OOD data.

**Limitations.** One concern might be the reliance on synthetically generated diffusion images and MLLM-extracted features for domain awareness. However, this is mitigated by: (1) the small size of the diffusion dataset (4096 samples), (2) images synthesized using generic, class-agnostic style prompts, and (3) the MLLM processing multiple style-consistent images, which focuses it on style over objects. Furthermore, DANN [16] and our ablations without disentanglement (Table 4), even with such data, fails to improve CLIP's OOD performance (Table 3).

The role of the MLLM may also be questionable, as LLaVA internally uses a CLIP-L encoder. However, LLaVA's poor zero-shot image classification performance (Table 4), a known issue attributed to MLLMs' improper alignment for classification [48], justifies not using it as a direct classifier. Instead, we use an MLLM because CLIP captures global information from images, which prioritizes overall style [49], making its representations suitable for domain-level information. The MLLM, with its language capabilities, is then able to explain the perceived domain styles into textual descriptions and provide informative hidden state representations.

Lastly, our unlearning strategy involves making DomainNet images and random noise indistinguishable, differing from the standard approach [15] where samples are typically mapped to known OOD data. This adaptation was necessary as CLIP's extensive web-scale pretraining makes finding truly unseen data challenging. Future work could explore more sophisticated unlearning methods for DG in-the-wild evaluation. Nevertheless, the significant degradation observed in zero-shot performance post-unlearning, and the fact that PEFT methods showed improvements on less OOD data but poorer performance on more OOD data, is evidence that our unlearning procedure functioned as intended.

# 6 Conclusion

In this work, we highlighted the potential limitations of current DG evaluation settings for foundation models like CLIP, which may not adequately test unseen data scenarios. We instead used a more challenging and comprehensive evaluation to simulate DG in-the-wild, with quantified OOD scores for target datasets, and an unlearning approach to further simulate unseen data. To address the challenges of DG in-the-wild, we introduced CLIP-DCA. Our method disentangles classification from domain-aware representations, motivated by the idea that while domain invariance is important for performance on unseen data, domain awareness is important to retain the vast pretrained knowledge of CLIP. Overall, our method significantly improves OOD robustness over existing baselines.

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
