# OpenReview forum: "Domain Generalization in-the-Wild: Disentangling Classification from Domain-Aware Representations"
_NeurIPS.cc/2025/Conference — Submitted to NeurIPS 2025_

### Official Review · Reviewer_CRMh · 2025-06-09

**Clarity:** 3
**Significance:** 2
**Originality:** 2
**Rating:** 3
**Confidence:** 3

**Summary:**

The paper proposes CLIP-DCA, a fine-tuning method that improves CLIP’s robustness to unseen domains by disentangling classification from domain-aware features. Using synthetic data and unlearning, it evaluates on diverse datasets and shows better OOD performance than standard baselines.

**Questions:**

* Does the added domain head and training complexity affect inference cost or latency?

* How is CLIP-DCA fundamentally different from existing disentanglement or domain-invariance methods?

* Which component of CLIP-DCA contributes most to performance gains? Can the disentanglement alone explain the improvements?

* See the weaknesses above

**Ethical Concerns:**

["NO or VERY MINOR ethics concerns only"]

**Final Justification:**

After carefully reading the rebuttal and other reviews I decided to keep my score.

**Limitations:**

yes

**Paper Formatting Concerns:**

No concerns

**Quality:**

3

**Strengths And Weaknesses:**

**Strengths**

* Clearly identifies an important problem: CLIP’s true generalization capabilities may be overestimated due to domain contamination.

* Introduces CLIP-DCA, a method that aims to disentangle classification from domain-awareness.

* Extensive empirical evaluation across various datasets.

* Novel use of synthetic domains generated via diffusion and MLLMs.
* Paper writing is clear and the framework figure is helpfull for  better understanding the approach

**Weaknesses**

* The central idea is not new. It builds on well-established ideas (domain heads, adversarial unlearning, disentanglement via decorrelation) without a significant leap in methodology.

* The architectural addition (a second linear head + contrastive and decorrelation losses) is incremental and lacks theoretical grounding.

* CLIP-DCA introduces several loss terms (C1 to C6), additional heads, synthetic data pipelines, and unlearning which complicate reproducibility.

---

> ### Author Rebuttal · Authors · 2025-07-31
>
> We thank the reviewer for their valuable feedback. We are encouraged that the reviewer found our problem setup “identifies an important problem” and that “the writing is clear.” We address the reviewer’s questions/concerns below.
>
> &nbsp;
>
> ## *Q1 Novelty*
> - Thank you for bringing up this important point. We agree that our method builds on well-established ideas. However, we would also like to point out that our novelty comes from our combination of different components to address the specific challenge of balancing awareness and invariance in CLIP. We present a different direction from the popular trend of using parameter-efficient finetuning (PEFT), where the original CLIP weights are minimally changed. We argue this trend arose due to the assumption of the inherent OOD robustness in CLIP. We challenge this assumption with our more challenging baseline (lines 102~128).
> - Additionally, we believe our paper is the first to suggest that balancing domain awareness and invariance are critical to truly improve DG performance for CLIP.
>
> &nbsp;
>
> ## *Q2 Contribution to performance gains*
> - As Table 4 shows, no one component contributes most to performance. The results show that both domain awareness AND disentangled invariance are necessary, and removing either one significantly harms performance. These  results support our main message that both awareness and invariance are needed to improve CLIP’s DG performance.
>
> &nbsp;
>
> ## *Q3 Inference Cost*
> - Thank you for asking this important question. We would like to re-emphasize that the added domain head and loss terms are training-only components. Inference latency and architecture are identical to the standard CLIP model (Figure 3).
> - Please note that the diffusion images and the MLLM text/hidden states are static. They were pre-computed prior to training, and they are only used during training. The external data and the additional head has no impact during inference.
>
> &nbsp;
>
> ## *Q4 Reproducibility*
> - Thank you for bringing up this important point. We provide the pseudo-code for creating diffusion images and creating captions and hidden states using an MLLM. Due to space restrictions, for other code, please refer to our comments in 9z2Q, Q5 for training and xm5e, Q7 for unlearning.
>
>
> ### **Diffusion images**
>
>
>     model_nf4 = SD3Transformer2DModel.from_pretrained(model_id, subfolder="transformer",
>         quantization_config=nf4_config, torch_dtype=torch.bfloat16)
>
>     pipeline = StableDiffusion3Pipeline.from_pretrained("stabilityai/stable-diffusion-3.5-medium",
>         transformer=model_nf4, torch_dtype=torch.bfloat16)
>     pipeline.safety_checker = None
>
>     device = "cuda"
>     pipeline = pipeline.to(device)
>
>     for style in domains:
>         for count in range(images_per_class):
>             images = pipeline(prompt=f"{style}", num_inference_steps=40, num_images_per_prompt=8).images
>             images.save()
>
>
>
> ### **MLLM**
>
>     processor = LlavaNextProcessor.from_pretrained("llava-hf/llama3-llava-next-8b-hf")
>
>
>     model = LlavaNextForConditionalGeneration.from_pretrained("llava-hf/llama3-llava-next-8b-hf", torch_dtype=torch.float16,
>                                                               attn_implementation='flash_attention_2', device_map='auto').eval()
>
>     mllm_outputs = {}
>     for domain in domains:
>         cur_images = images[domain]
>         conversation = [{"role": "user", "content": [
>                           {"type": "text", "text": f"... Describe the aspects of the style that applies regardless of category. ... "},
>                           *[copy.deepcopy({"type": "image"}) for _ in range(len(cur_images))]]}]
>
>
>         prompt = processor.apply_chat_template([conversation], add_generation_prompt=True)
>         inputs = processor(images=cur_images, text=prompt, return_tensors="pt").to(model.device)
>         outputs_hidden = model(**inputs, output_hidden_states=True, return_dict=True)
>         hiddens_states = outputs_hidden.hidden_states[-1].mean(dim=1).detach().cpu()
>
>
>         generate_ids = model.generate(**inputs, max_new_tokens=150)
>         outputs_text = processor.batch_decode(generate_ids, skip_special_tokens=True, clean_up_tokenization_spaces=False)
>
>         outputs_text = [output.split('assistant')[-1].replace("\n", "").replace('-', '').replace('*', '') for output in outputs_text]
>
>         mllm_outputs[domain] = {'hidden_states': np.array(hiddens_states).tolist(), 'text' outputs_text}
>
>
> &nbsp;
>
> Thank you again for your feedback and please feel free to ask any follow up questions or clarify any other concern you may have.

---

### Official Review · Reviewer_xm5e · 2025-06-26

**Clarity:** 3
**Significance:** 3
**Originality:** 3
**Rating:** 4
**Confidence:** 4

**Summary:**

This paper investigates in-the-wild domain generalization (DG) with CLIP. The main contributions are two-fold. (1) To assess the performance of CLIP on genuinely unseen data, this paper adopts unlearning to exclude potential target-like pretrained knowledge in the model. (2) A CLIP-DCA method is proposed to enhance DG aided by pretrained MLLMs and diffusion models.

**Questions:**

Please refer to the weaknesses

**Ethical Concerns:**

["NO or VERY MINOR ethics concerns only"]

**Final Justification:**

Thanks for the rebuttal. Most of my concerns have been resolved. While there might be slight unfairness for introducing external knowledge, the main contributions in methodology and problem setting in this paper are solid and appreciated. Therefore I lean towards acceptance.

**Limitations:**

yes

**Quality:**

3

**Strengths And Weaknesses:**

Strengths:
- The investigated ‘genuine OOD evaluation’ of CLIP is reasonable and appreciated.
- The paper is overall well-structured and easy to follow.

Weaknesses:
My major concerns focus on the Experiments, as detailed below:
- Clarity of experiments can be improved. (1) The unlearning algorithm in Sec. 4.3 is abstract and takes efforts to understand. It is suggested that framework figures and loss equations be added to clarify such process. (2) All result figures in the paper need further explanations. For example, in Fig.1, what are the dots and how are the lines obtained? How to compute the line results in Fig.8 and 9 also need clear explanations. The error bounds should be included. (3) In Fig.8, CoOp and CLIP-Adapter overlaps with each other, and they are not affected by OOD score. What is the reason?
- Compared baselines are incomplete. The compared baselines are mostly robust fine-tuning methods, while there are lots of recent CLIP-based DG methods [1-3]. How do they perform under these experimental settings? In addition, DANN is listed as a baseline. As I understand it, DANN is designed for unsupervised domain adaptation, how is it applied for DG?
- Unfair comparisons. The proposed method introduces much external knowledge, i.e., the pretrained MLLM for feature alignment, and diffusion model to generate extra training samples. This is unfair for other compared methods.
- This paper introduces a new setting with many modifications to the public CLIP and introduces many external data and models in the proposed method. As the code is not provided, and some details in training and data splits are omitted (e.g., unlearning process), I have concerns on the reproducibility of the method and results in this paper.

[1] Addepalli S, et al. Leveraging vision-language models for improving domain generalization in image classification, 2024.
[2] Chen Z, et al. Practicaldg: Perturbation distillation on vision-language models for hybrid domain generalization, 2024.
[3] Singha M, et al. Unknown Prompt the only Lacuna: Unveiling CLIP's Potential for Open Domain Generalization, 2024.

---

> ### Author Rebuttal · Authors · 2025-07-31
>
> We thank the reviewer for the valuable constructive criticism. We are encouraged that the reviewer found our problem setup “reasonable and appreciated.” Regarding the reviewer’s major concern about reproducibility, we provide pseudo-code, given we are disallowed from posting links. Please let us know if you would like any more details on any component. We address the reviewer’s other questions/concerns below.
>
> &nbsp;
>
> ## *Q1 Experimental Clarity*
> - Thank you very much for your suggestions to improve the clarity of our paper.
>     1) To clarify our unlearning algorithm, which is very similar to a recent unlearning technique (ref. 15 in main text), we will add a figure (similar to Figure 1 in ref. 15) to more clearly explain the framework and keep our paper more self-contained. We also provide a pseudocode in Q7. In short:
> 	    - To prevent forgetting of its original knowledge, the GCC dataset is used as a proxy of CLIP’s pretraining data and is used to train CLIP.
> 	    - Simultaneously, a binary classifier classifies DomainNet images and random noise. The model is penalized if the classifier is able to classify domains from random noise (using a gradient reversal layer). The goal is to make domain features indistinguishable from random noise while retaining its pretraining knowledge.
>     2) We will explicitly define figure elements. In figure 1, each dot is a dataset, and the lines are regression lines to best fit the dots. We will also add error bounds to regression plots (Figs. 8/9).
>     3) CoOp/Adapter lines in Fig 8 are similarly flat because as parameter-efficient finetuning (PEFT) methods, they freeze ~99% of CLIP's weights, making their performance very close to the zero-shot baseline (lines 314\~325).
>
>
> &nbsp;
>
> ## *Q2 Baselines*
> - Thank you for bringing up this important point. We will work towards providing more methods for comparison. However, please note that the suggested [1-2] are distilling methods that train a traditional image classification model (eg. CNN). They cannot be used in a cross-dataset setup where the target dataset has different class names. While [3] can be tested in a cross-dataset setup, it seems there is label leakage during inference in their **official repository**. A github issue from 2024 (Issue #5) also documented the label leakage problem. When we run their code on ImageNet, it achieves **~95% on the ImageNet validation set**, which would be better than the current SOTA. Due to time constraints, we were unable to resolve the issue.
>
>
>
>    | | INet v1 | \-V2 | \-Sketch | \-A | \-R | Avg. on 33 |
>    | ---------------- | ---- | ---- | --------- | ---- | ---- | --------- |
>    | Zeroshot | 46.0 | 40.4 | 27.4 | 15.1 | 51.6 | 45.5 |
>    | Regular Finetune | 69.8 | 58.4 | 34.7 | 15.0 | 52.6 | 43.6 |
>    | CoOp | 53.3 | 46.2 | 29.1 | 16.1 | 52.8 | 45.5 |
>    | MMA [4] | 71.7 | 60.0 | 36.1 | 7.8 | 37.0 | 46.3 |
>    | LwEIB [5] | 53.8 | 46.7 | 30.4 | 16.3 | 54.1 | 46.6 |
>    | CLIP-DCA (Ours) | 75.1 | 63.9 | 42.2 | 22.9 | 62.2 | 52.1 |
>
>
> - Instead, we looked at two other recent studies that improve CLIP’s domain generalization performance. We used their official public repositories. These methods also test with a smaller cross-dataset [4-5]. We find that MMA [4] achieves similar results for the in-distribution Imagenet v1 (c.f. Table 3 in MMA paper), but the OOD performance (v2, Sketch, A, R) is generally lower. For LwEIB [5], the overall robustness is very slightly higher than MMA, as seen by the slightly higher average across 33 datasets.
> - These results align with what we have presented for the more traditional PEFT methods (like CoOp). While they provide a small increase in performance for in-distribution data, they do not consistently provide improvements on OOD data. They are also limited by their constraint to only change a small percentage of the parameters.
> - We argue that the underlying assumption that CLIP is robust to OOD data may not actually be consistent in a truly in-the-wild setting (lines 123\~128).
>
> &nbsp;
>
>
> ## *Q3 DANN as a DG Baseline*
> - It is true that DANN was proposed as a domain adaptation technique. However, the core mechanism of DANN, adversarially learning domain-invariant features, is also a foundational idea in DG. The goal of DG is to train a model to be robust to domain shifts, and enforcing domain invariance is a primary strategy to achieve this, as mentioned in an influential DG paper [6]. Learning domain invariance is also arguably the most common approach across the DG field [7]. As such, many papers across recent years (2022\~2025) still use DANN as a DG baseline, especially when comparing methods that learn domain invariance [8-12].  While we do mention this connection briefly (lines 90\~101), we will revise to more clearly explain this connection.
>
> &nbsp;
>
> ## *Q4 Unfair Comparison (External Data)*
> - Thank you for bringing up this valid concern. We also discussed it as our first limitation (lines 400~405). The main evidence to support our claims is that even using the same data, learning domain invariant representations and domain aware representations separately is ineffective.
> 	- DANN (Table 2), which also uses the new data, shows worse performance compared to other methods even with its access to the diverse domain data.
> 	- Additionally, our ablations (Table 4), which shows the setting where only domain awareness is learned is also ineffective.
> 	- We argue that the combination of awareness and disentanglement is what makes our method truly effective.
> - We additionally point out in the limitations that the dataset is very small (4096 images), and that the images are class-agnostic.
>
>
> &nbsp;
>
> ## *Q5 Model Modification*
> - We respectfully disagree that many modifications have been made. Our new losses are mostly based on the outputs of the original CLIP architecture. We only add two linear layers, the domain head and the MLLM projector, which are only during training. Due to space constraints, for our training pseudocode, please see our comments on Reviewer 9z2Q (Q5). Please also note that the inference architecture is identical to the original CLIP, with no added complexity or cost (Figure 3).
>
> &nbsp;
>
> ## *Q6 Details in data splits*
> - We are unfortunately not able to send an anonymous link to our code. We provide a simple pseudocode here. Please feel free to clarify any source of unclarity. We will happily provide more code during our discussion phase.
> - We only train on ImageNet, GCC, and Diffusion images. We test on other, completely separate, datasets.
> - Imagenet: official training set on huggingface
>
>
>         from datasets import load_dataset
>         imagenet_dataset = load_dataset('imagenet-1k', split='train')
>
> - GCC data: public huggingface dataset
>
>         liuhaotian/LLaVA-CC3M-Pretrain-595K
> - We used the huggingface library for the Diffusion model and the MLLM.
>
>         Diffusion: SD3Transformer2DModel, stabilityai/stable-diffusion-3.5-medium
>         MLLM: LlavaNextForConditionalGeneration, llava-hf/llama3-llava-next-8b-hf
>
>
> &nbsp;
>
> ## *Q7 Unlearning pseudocode (explanation in Section 4.3)*
>
>         # Please note that logit scales (temperature term) was omitted for simplicity
>         discriminator = nn.Linear()
>
>         # Following [7]
>         p = float(batch_idx + start_steps) / total_steps
>         alpha = 2. / (1. + np.exp(-10 * p)) - 1
>
>         def classification_loss(image, text):
>             logits_per_image = image @ text.T
>             logits_per_text = text @ image.T
>             cur_labels = torch.arange(len(image), device=device, dtype=torch.long)
>             return (F.cross_entropy(logits_per_image, cur_labels)+F.cross_entropy(logits_per_text, cur_labels))/2
>
>
>         # 1. GCC datasets
>         gcc_images, gcc_captions = gcc_batch
>         _, image_embeddings, text_embeddings = clip_model(gcc_images, gcc_captions)
>         gcc_loss = classification_loss(image_embeddings, text_embeddings)
>
>         # 2. DomainNet
>         dn_images, _ = gcc_batch
>         random_noise = torch.randn((len(dn_images), 3, 224, 224), requires_grad=True, device=device)
>         penultimate_image_embeddings, _ = clip_model.encode_image(torch.cat((dn_images, random_noise), dim=0))
>
>         x = discriminator(penultimate_image_embeddings.clone(), alpha)
>         domain_targets = torch.cat((torch.zeros(len(dn_images)),torch.ones(len(dn_images))), dim=0)
>         domainnet_loss = F.cross_entropy(x, domain_targets)
>
>         # 3. Final loss
>         loss = gcc_loss + domainnet_loss
>
> &nbsp;
>
> References:
>
> ##### [1] Leveraging vision-language models for improving domain generalization in image classification, 2024.
>
> ##### [2] Practicaldg: Perturbation distillation on vision-language models for hybrid domain generalization, 2024.
>
> ##### [3] Unknown Prompt the only Lacuna: Unveiling CLIP's Potential for Open Domain Generalization, 2024.
>
> ##### [4] MMA: Multi-Modal Adapter for Vision-Language Models (CVPR 2024)
>
> ##### [5] Learning with Enriched Inductive Biases for Vision-Language Models (IJCV 2025)
>
> ##### [6] In search of lost domain generalization (ICLR 2021)
>
> ##### [7] Wang, Jindong, et al. "Generalizing to unseen domains: A survey on domain generalization." (2022)
>
> ##### [8] Cha, Junbum, et al. "Domain generalization by mutual-information regularization with pre-trained models." (ECCV 2022)
>
> ##### [9] Li, Aodi, et al. "Learning common and specific visual prompts for domain generalization." Proceedings of the Asian conference on computer vision. 2022.
>
> ##### [10] Cho, Junhyeong, et al. "Promptstyler: Prompt-driven style generation for source-free domain generalization." (CVPR 2023)
>
> ##### [11] Disentangled prompt representation for domain generalization (CVPR 2024)
>
> ##### [12] Selective unlearning via representation erasure using domain adversarial training (ICLR 2025)

---

> > ### Comment · Reviewer_xm5e · 2025-08-09
> >
> > Thanks for the rebuttal. Most of my concerns have been resolved. While there might be slight unfairness for introducing external knowledge, the main contributions in methodology and problem setting in this paper are solid and appreciated. Therefore I lean towards acceptance.

---

> ### Author Response · Authors · 2025-08-06
>
> Thank you again for your constructive feedback.
>
> As we near the end of our discussion phase, we would like to carefully follow up to check whether our rebuttal has answered your questions.
>
> We would like to point out that our additional baselines attempt to directly address your main concern regarding our experimentation.
>
> Please let us know if there is anything we can further clarify or if there are concerns that have not been properly addressed.

---

> > ### Author Response · Authors · 2025-08-08
> >
> > Dear reviewer xm5e,
> >
> > We're following up one last time before our discussion phase ends tomorrow.
> >
> > Please let us know if we can clarify anything, especially regarding your main concerns about our experiments.
> >
> > Thank you,
> >
> > Authors

---

### Official Review · Reviewer_9z2Q · 2025-06-30

**Clarity:** 3
**Significance:** 3
**Originality:** 3
**Rating:** 5
**Confidence:** 4

**Summary:**

This paper addresses domain generalization in-the-wild, a significant and novel problem with huge implications. The authors hypothesize that domain awareness can lead to better generalization without compromising model performance—an intuition that is well justified. They quantify “wildness” across 33 datasets, simulate unseen domains via an unlearning procedure on a proxy dataset (GCC), and introduce CLIP-DCA, which disentangles classification from domain-aware representations in CLIP finetuning. Empirical results show very strong gains on OOD benchmarks.

**Questions:**

* **LLM Choice:** How would your results change if you used a more powerful LLM (e.g., GPT-4o) for domain generation? Could richer prompts further improve OOD gains?
* **Unknown Domains:** Many generated domains may already appear in the LLM’s training data—so the model can simply reproduce them (e.g., “charcoal sketch drawing,” closely matching the Sketch domain in PACS). As a result, these experiments may not fully reflect CLIP-DCA’s performance on truly novel domains (not seen by the LLM or mixture of different domains). How might the authors address this limitation? Does awareness of a diverse set of domains enhance performance on other unseen, unknown domains?
* **Unlearning Validation:** Report zero-shot CLIP accuracy on DomainNet (and ImageNet or a similar standard dataset) immediately before and after your unlearning procedure to demonstrate its effectiveness.

**Ethical Concerns:**

["NO or VERY MINOR ethics concerns only"]

**Final Justification:**

The authors have addressed my major concerns about unlearning performance and the impact of larger LLMs through a comprehensive discussion and new experiments. Hence, I am now recommending this paper for acceptance.

**Limitations:**

The authors include a dedicated Limitations section that thoughtfully explores various potential shortcomings of their approach and explains the nuances behind key design choices.

**Quality:**

3

**Strengths And Weaknesses:**

## Strengths:
*  The problem is significant and novel with huge implications.
* The hypothesis that domain awareness improves generalization without performance compromise is intuitive and well-justified.
* The loss formulation is explained very well.
* Supplementary material contains all details required for replication.
* The method demonstrates very strong results, especially in Fig. 9 and Table 2.

## Weaknesses
* The training pipeline in Section 3 is rather vague and needs further elaboration.
* Reliance on a single small LLM (8B) for domain generation may bottleneck domain diversity; impact of more powerful LLMs (e.g., GPT-4o) is unexplored.
* LLM‐generated domains may overlap with test domains because LLMs are often trained on dataset metadata and the corresponding literature; as a result, the method’s robustness to truly novel domains remains untested.
* Fig. 6 lacks a regression line to illustrate the OOD‐score vs. accuracy relationship (Sec. 4.2).
* Effectiveness and the potential negative impact of the unlearning process is not fully demonstrated: zero-shot CLIP performance before/after unlearning on DomainNet and ImageNet (or similar dataset) is missing.

---

> ### Author Rebuttal · Authors · 2025-07-31
>
> Thank you for your thought-provoking comments. We are encouraged that the reviewer found our work to tackle a "significant and novel problem with huge implications." We address the reviewer’s questions/concerns below.
>
> &nbsp;
>
> ## *Q1 LLM Choice*
>
>
> | | INet v1 | \-V2 | \-Sketch | \-A | \-R | Avg. on 33 |
> | --------------------- | ------- | ---- | -------- | ---- | ---- | ---------- |
> | Zeroshot | 46.0 | 40.4 | 27.4 | 15.1 | 51.6 | 45.5 |
> | Regular Finetune | 69.8 | 58.4 | 34.7 | 15.0 | 52.6 | 43.6 |
> | Ours (Gemini 2.5 Pro) | 76.1 | 64.9 | 42.7 | 22.9 | 61.9 | 52.5 |
> | Ours (Llava-8B) | 75.1 | 63.9 | 42.2 | 22.9 | 62.2 | 52.1 |
>
>
> - Thank you for this great suggestion. We used Gemini 2.5 Pro to test a much larger MLLM. Gemini did provide a slight improvement in average performance that mainly came from the less OOD datasets. Due to the limited time, we used only our default hyperparameters.
> - The small difference between Gemini and the smaller MLLM provides further evidence that CLIP is not simply benefitting from larger MLLMs’ capacity, which is intuitive considering that the diffusion dataset is only 4096 samples (8 images for 512 domains). We believe the more important aspect of our work is the disentanglement between the domain and class head that is designed to encourage the model to effectively learn domain invariant representations.
>
> &nbsp;
>
> ## *Q2 Unknown domains*
> - Thank you for bringing up this critical point. This is a valid concern, and we also mention it as our first limitation (lines 400~405).
> - However, the main evidence to support our claims is that even using the same data, learning domain invariant representations and domain aware representations separately is ineffective.
>     - DANN (Table 2), which learns only domain invariant representations, shows worse performance compared to other methods even though it has access to the diverse domain data.
>     - Our ablations (Table 4), which shows the setting where only domain awareness is learned, is also ineffective.
>     - It is the combination of domain invariance and domain awareness  through disentanglement that  makes our method truly effective.
> We additionally point out in the limitations that the dataset is quite small (4096 images), and that the images are class-agnostic.
>
>
>
> &nbsp;
>
>
> ## *Q3 Unlearning validation and effectiveness*
> - Thank you for this excellent suggestion. We agree that demonstrating the direct effect of unlearning is important.
> - To clarify our method, the unlearning process requires simultaneous retention training on the GCC dataset to prevent catastrophic forgetting. Because of this, retention training is a requirement for the unlearning framework. Please note that we follow [1] (ref. 15 in main text).
> - As such, Table 1 presents:
>     1) Zeroshot: Performance **immediately before** unlearning
>     2) Regular fine tuning: Baseline performance after fine tuning on GCC (retention only)
>     3) Unlearning: Performance **immediately after** our full unlearning procedure (simultaneous retention on GCC and unlearning on DomainNet).
> - The effectiveness of the unlearning is shown more clearly when taking into account Figure 9, which shows that PEFT performance degrades for more OOD data, which is intuitive. Whereas before unlearning, PEFT methods would improve performance regardless of OOD score supporting the argument for domain contamination (Figure 8).
>
> &nbsp;
>
> ## *Q4 Regression line (Fig 6)*
> - We will add a regression line to Figure 6 to better visualize the relationship between OOD score and accuracy. We will also add error bounds for all regression lines. Thank you for the suggestion.
>
> &nbsp;
>
> ## *Q5 Training pipeline elaboration*
> - Thank you for your suggestion. We will add an algorithms/pseudocode section for our training setup in the appendix, which we provide below.
>
>         # Please note that logit scales (temperature term) was omitted for simplicity
>         def classification_loss(image, text):
>             logits_per_image = image @ text.T
>             logits_per_text = text @ image.T
>             cur_labels = torch.arange(len(image), device=device, dtype=torch.long)
>             return (F.cross_entropy(logits_per_image, cur_labels)+F.cross_entropy(logits_per_text, cur_labels))/2
>
>         def disentangle_loss(x,y):
>             x = (x - x.mean(0)) / (x.std(0)+1e-8)
>             y = (y - y.mean(0)) / (y.std(0)+1e-8)
>             crossCorMat = (x@y.T) / len(x)
>             return torch.diagonal(crossCorMat).pow(2).sum()
>
>         # Equivalent to classification loss
>         def MLLM_agreement_loss(x_, y_):
>             x_per_y = x_ @ y_.T
>             y_per_x = y_ @ x_.T
>             cur_labels = torch.arange(len(x_), device=device, dtype=torch.long)
>             return (F.cross_entropy(x_per_y, cur_labels) + F.cross_entropy(y_per_x, cur_labels) ) / 2
>
>         #0. Architectural addition
>         domain_head = nn.Linear()
>         MLLM_projector = nn.Linear()
>
>         #1. Diffusion
>         d_images, d_hidden, d_text = d_batch
>
>         penultimate_image_embeddings, class_image_embeddings, text_embeddings = clip_model(d_images, d_text)
>         domain_image_embeddings = domain_head(penultimate_image_embeddings.clone())
>         domain_mllm_embeddings = MLLM_projector(d_hidden)
>
>         d_agreement_loss = classification_loss(domain_image_embeddings.clone(), text_embeddings.clone())
>         d_MLLM_loss = MLLM_agreement_loss(domain_mllm_embeddings.clone(), text_embeddings.clone())
>         d_dist_loss = (disentangle_loss(class_image_embeddings.clone(), domain_image_embeddings.clone())
>                      + disentangle_loss(class_image_embeddings.clone(), text_embeddings.clone())) / 2
>
>
>         #2. Source images
>         s_images, s_text = s_batch
>         penultimate_image_embeddings, class_image_embeddings, text_embeddings = clip_model(s_images, s_text)
>         domain_image_embeddings = domain_head(penultimate_image_embeddings.clone())
>
>         s_class_loss = classification_loss(class_image_embeddings.clone(), text_embeddings.clone())
>         s_dist_loss = disentangle_loss(class_image_embeddings.clone(), domain_image_embeddings.clone())
>
>
>         #3. Final loss
>         loss = d_agreement_loss+d_MLLM_loss+d_dist_loss+s_class_loss+s_dist_loss
>
>
> - Note that inference is equivalent to the original CLIP model (Figure 3).
>
>         image_features = clip_model.encode_image(image)
>         text_features = clip_model.encode_text(text)
>         image_features /= image_features.norm(dim=-1, keepdim=True)
>         text_features /= text_features.norm(dim=-1, keepdim=True)
>
>         probabilities = (100.0 * image_features @ text_features.T).softmax(dim=-1)
>
> &nbsp;
>
> Thank you once again for your comments, and please feel free to clarify any of your questions/concerns if it has not been fully answered.
>
> &nbsp;
>
> References:
>
> ##### [1] Sepahvand, Nazanin Mohammadi, et al. "Selective unlearning via representation erasure using domain adversarial training." The Thirteenth International Conference on Learning Representations. 2025.

---

> > ### Comment · Reviewer_9z2Q · 2025-08-05
> >
> > I would like to thank the authors for their response. I have studied their rebuttal, as well as the other reviewers’ comments, carefully. The authors have addressed most of my major concerns; hence, I am increasing my initial score.

---

> > > ### Author Response · Authors · 2025-08-06
> > >
> > > Thank you very much once again for your feedback and questions.
> > >
> > > Please let us know if you have any follow up questions before the end of the discussion phase.

---

### Official Review · Reviewer_MyYa · 2025-07-02

**Clarity:** 4
**Significance:** 3
**Originality:** 3
**Rating:** 5
**Confidence:** 4

**Summary:**

This paper addresses the problem of domain generalization (DG) in foundational models trained on internet-scale data. It begins by motivating the challenge of evaluating DG in such models, particularly for truly unseen domains, and supports this with evidence from a recent study. The authors then propose two approaches to simulate out-of-distribution scenarios for foundational models: (1) evaluating on 33 challenging datasets, and (2) applying the concept of unlearning. Building on this, the paper introduces a method that combines the traditional DG objective of learning domain-invariant features with a novel focus on learning domain-aware representations. The goal of domain-aware learning is to retain useful features when fine-tuning on new domain data. The proposed approach leverages domain descriptions generated by a multimodal LLM, domain-specific images synthesized by a diffusion model, and combines these with existing DG datasets. These components are used to fine-tune image and text encoders in a way that promotes domain awareness, domain invariance, and class discriminability.

**Questions:**

It would be helpful for the paper to clarify how the disentanglement loss defined as the squared sum of the diagonal of the cross-correlation matrix, relates to the conceptual goal of disentanglement. Specifically, how does this formulation encourage the separation of factors such as domain and class information, and how does it support the overall objective of domain awareness? Similarly, a more explicit explanation of what the disentanglement losses aim to achieve would be valuable. Do it seek to learn representations where class and domain-related features are disentangled? If so, how is this connected to the broader goals of learning domain-aware and domain-invariant features?

Regarding the use of Stable Diffusion to generate domain-specific images without class labels, it would be useful to expand on how accurate or effective these generated images are in representing the intended domains. Including example images and discussing specific cases where Stable Diffusion performs well, and where it fails, would greatly strengthen the reader’s understanding. For instance, in scenarios involving subtle environmental shifts, such as those in the TerraIncognita dataset where only camera IDs are available, it’s unclear whether diffusion models can generate meaningful variations. It would also be interesting to see examples of the style descriptions used and identify any consistent failure cases, particularly when domains are abstract or under-specified.

The paper employs domain unlearning as way to simulate trueuly out-of-distribution examples for foundational models, using DomainNet as their dataset. While this approach is intuitively appealing, it raises some concerns. DomainNet shares semantic overlap with datasets like ImageNet. For example, both contain "bus" classes but the domains in DomainNet differ visually (e.g., sketch vs. photo). How does the unlearning process avoid removing general class-related features while targeting domain-specific ones? More broadly, how can one ensure that domain-specific variations (e.g., sketch-specific features of a bus) are unlearned without degrading class-level semantics? Some discussion of the general challenges in domain unlearning and potential strategies to mitigate these risks would be welcome.

It would also be interesting to evaluate how the fine-tuning methods, such as proposed method or PEFT-based approaches, affect the original performance of CLIP. For example, does zero-shot performance on ImageNet degrade after domain-aware fine-tuning for DG? Including such an analysis could provide insights into trade-offs between generalization to new domains and retention of performance on original tasks.

As a minor suggestion, the term "MLLM" (multimodal large language model) should be clearly defined before its first use to ensure clarity for all readers.

**Ethical Concerns:**

["NO or VERY MINOR ethics concerns only"]

**Final Justification:**

Overall, it is good paper. I have had some conners regarding clarity and contribution of certain parts of the proposed method. However, author's response have cleared my concern. I would suggest authors to add these modifications in the final version of the paper. I will keep my original rating of 5.

**Limitations:**

Limitations are adequately discussed in the paper. However, presentation of samples gnereated by diffusion models and MLLM could improve the paper.

**Quality:**

3

**Strengths And Weaknesses:**

### Strengths

The paper is well‑written and easy to follow, presenting its ideas and methodology with clarity. It tackles a timely and under‑explored challenge of evaluating domain‑generalization performance in large foundational models. A particularly noteworthy contribution is the introduction of domain awareness, which complements the traditional focus on domain invariance and broadens the conceptual toolkit for DG research. The authors design creative, well‑motivated experiments, including the use of challenging datasets and unlearning‑based strategies, to probe their method’s effectiveness. These experiments are comprehensive, offer strong comparative baselines, and are accompanied by thorough analyses. Finally, the paper discusses the limitations of the proposed approach.

### Weankesses

While the paper is strong overall, the Methods section feels somewhat limited in scope and lacks clear connections between some of its components and the broader goals of the work. For example, the role of disentanglement losses is not clearly linked, either intuitively or mathematically, to the objectives of achieving domain awareness or invariance. A more explicit explanation of how these losses contribute to the model’s behavior would strengthen the paper.

Additionally, while the use of multimodal large language models (MLLMs) and diffusion models is discussed both in the Methods and Limitations sections, the descriptions remain relatively high level. A more comprehensive and detailed explanation of how these components are integrated, and why they are effective in this context, would enhance the reader’s understanding and confidence in the approach.

In the experimental setup, domain adversarial training is employed to "unlearn" certain domains. Although this is intuitively reasonable, the paper would benefit from a brief discussion of the broader field of machine unlearning. Highlighting connections to prior work or established principles in unlearning could strengthen the conceptual grounding and clarify how this aspect fits into the overall domain generalization framework.

---

> ### Author Rebuttal · Authors · 2025-07-31
>
> We thank the reviewer for the thoughtful, detailed, and overall positive comments. We are also encouraged that the reviewer found our work to tackle a "timely and under-explored challenge." We address the reviewer’s questions/concerns below.
>
>
> &nbsp;
>
> ## *Q1 Role of Disentanglement Loss*
> - Thank you for the insightful question. We agree that the connection between the disentanglement loss and our goals of domain awareness and invariance can be made more explicit.
> - Our losses encourages the model to learn two distinct types of representations from a shared encoder:
>     - Domain-aware representations from the domain head, which should capture domain-specific features regardless of its class.
>     - Domain-invariant representations from the class head, which should capture the class-specific features regardless of its domain.
> - The role of the disentanglement loss is to enforce this separation. The assumption is that if the two representations are truly disentangled, the class head features for a given sample should be statistically independent from features learned by the domain head for that same sample.
> - We achieve this by minimizing the correlation between the class and domain embeddings for each sample in a batch. The loss is formulated as the squared sum of the diagonal of the cross-correlation matrix between the batch-normalized class embeddings (x) and domain embeddings (y).
>
>
>         x = (x - x.mean(0)) / x.std(0)
>         y = (y - y.mean(0)) / y.std(0)
>
>         crossCorMat = (x@y.T) / len(x)
>         loss = torch.diagonal(crossCorMat).pow(2).sum()
>
> - The disentanglement encourages the class head to find representations that are predictive of the class label without using features that are also useful for predicting the domain. Conversely, it encourages the domain head to focus only on domain-specific information, as any shared information with the class head is penalized. While there are other ways to achieve disentanglement, such as estimations for mutual information, we draw partial motivation from the simplicity of Barlow Twins [1] and VicReg [2] (self-supervised representation learning) by using the cross-correlation matrix between two vector embeddings.
>
> &nbsp;
>
>
> ## *Q2 Role of MLLM & Diffusion images*
> - We will define MLLM on its first use. Thank you for your suggestion.
> - The goal of using the MLLM and diffusion images is to encourage the model to learn domain awareness. As mentioned in Sec 3.2, most datasets lack explicit multiple source domains, and ones that do only have a few discrete domains, which motivated us to create our own dataset. The MLLM then effectively creates domain descriptions.
> - We will also add to the appendix qualitative examples of the generated diffusion images and generated MLLM descriptions (successes and failures) to demonstrate they provide diverse, class-agnostic style signals. While we aren’t allowed to post figures in our rebuttal, we do have a list of the 512 domains used in our appendix.
>
>
> &nbsp;
>
> ## *Q3 Background of unlearning and connection to DG*
> - Thank you for your suggestion, we have written a brief outline for the background here. Feel free to suggest any changes.
>
>     - Initially, unlearning was developed to support the right to be forgotten – to make the model behave as it had never seen certain data [3].
>     - Unlearning has been used in a wider range of fields such as representation learning. A particularly similar example is the field of domain adaptation (DA), where forgetting is used to align the model to the target domain [4].
>     - Since DG assumes no access to the target domain, and since we hypothesize that domain contamination is an issue due to the massive pretraining data, we instead use unlearning as a proxy to get to a state in which the model has not already seen the target domain.
>     - While exact unlearning (training from scratch without target domain) would be optimal, computation cost led us to use a recent approximate unlearning technique that uses Domain Aversarial Training (DANN) [5].
>
> &nbsp;
>
>
> ## *Q4 Unlearning concerns*
> - In the unlearning approach, the domains of DomainNet are used for unlearning. Simultaneously, we are also training the model using the GCC dataset (a proxy for CLIP's pre-training data) to prevent the catastrophic forgetting of general class features. The idea is that CLIP unlearns the ability to classify DomainNet’s domains while retaining its original information using the GCC dataset. To add clarity to this point and section 4.3, we will add a figure to show this setup, similar to Figure 1 used in [4].
>
>
> &nbsp;
>
> ## *Q5 Impact on Original Performance*
> - We assume that the reviewer is referring to the performance before and after fine tuning? Table 2 and Figure 9 shows the zeroshot results on ImageNet. They also show the results after fine tuning using various methods. In short, fine tuning methods improve similarly on in-distribution data, while our method degrades significantly less on highly OOD data, demonstrating a better trade-off.
>
> &nbsp;
>
> Thank you once again for your comments, and please feel free to clarify any of your questions/concerns if it has not been fully answered.
>
> &nbsp;
>
> References:
>
> ##### [1] Zbontar, Jure, et al. "Barlow twins: Self-supervised learning via redundancy reduction." International conference on machine learning. PMLR, 2021.
>
> ##### [2] Bardes, Adrien, Jean Ponce, and Yann LeCun. "Vicreg: Variance-invariance-covariance regularization for self-supervised learning." arXiv preprint arXiv:2105.04906 (2021).
>
> ##### [3] Shaik, Thanveer, et al. "Exploring the landscape of machine unlearning: A comprehensive survey and taxonomy." IEEE Transactions on Neural Networks and Learning Systems (2024).
>
> ##### [4] Basak, Hritam, and Zhaozheng Yin. "Forget More to Learn More: Domain-Specific Feature Unlearning for Semi-supervised and Unsupervised Domain Adaptation." European Conference on Computer Vision. Cham: Springer Nature Switzerland, 2024.
>
> ##### [5] Sepahvand, Nazanin Mohammadi, et al. "Selective unlearning via representation erasure using domain adversarial training." The Thirteenth International Conference on Learning Representations. 2025.

---

> > ### Author Response · Authors · 2025-08-06
> >
> > Thank you again for your positive review of our paper.
> >
> > We would like to follow up to check whether our rebuttal has answered all your questions.
> >
> > Please let us know if there is anything we can further clarify.

---

> > ### Comment · Reviewer_MyYa · 2025-08-06
> >
> > Overall, it is good paper. I have had some conners regarding clarity and contribution of certain parts of the proposed method. However, author's response have cleared my concern. I would suggest authors to add these modifications in the final version of the paper. I will keep my original rating of 5.

---

### Note · Authors · 2025-08-11

Dear AC,

Thank you for overseeing our paper! We are grateful for the reviewers' constructive feedback, which has strengthened our paper.

We are encouraged that three reviewers now support our paper's acceptance. Through our discussion, we addressed major concerns, leading Reviewer 9z2Q to raise their score to Accept (5) and Reviewer xm5e to raise their score from Borderline Reject (3) to "leaning towards acceptance."

Reviewers who engaged in discussion mentioned that our paper tackles a "timely and under-explored challenge" (MyYa)  on a "significant and novel problem with huge implications" (9z2Q), and that our "main contributions in methodology and problem setting... are solid and appreciated" (xm5e).

Their initial concerns regarding our experimental setup, baselines, and clarity were resolved through the clarifications in our rebuttal.



Reviewer CRMh maintained a Borderline Reject (3) score, citing novelty as the main concern. While we provided a detailed rebuttal, we were unfortunately unable to engage in further discussion to resolve this point.
We respectfully argue that this view overlooks our core contributions:
- We believe we are the first to demonstrate that balancing domain awareness and invariance is critical for improving DG in large foundation models like CLIP.
- We propose a novel method that diverges from standard PEFT methods by challenging the assumption of CLIP's inherent OOD robustness and showing the benefits of end-to-end fine-tuning.

In summary, we believe our work, recognized by reviewers as addressing a significant and impactful problem, represents a valuable and novel contribution. We hope the AC will consider the supportive reviews and our detailed rebuttal.

Thank you,

Authors

---

### Decision · Program_Chairs · 2025-09-17

**Decision:**

Reject

**Comment:**

This paper was reviewed by four experts who provided diverging recommendations, which generally tended towards recommending acceptance.  This paper observes that many domain generalization benchmarks that use large models like CLIP may be affected by data leakage, i.e., where during pretraining they may have seem the domains that DG benchmarks evaluate.  Thus, methods tested on them may not be measuring DG performance, an issue when trying to push evaluate methods for their DG ability.  The authors provide two mechanisms for addressing this.  First, they produce a benchmark that takes a model like CLIP, unlearns features that are specific to domains evaluated within the benchmark, thereby ensuring that they are OOD.  Second, they use a generative model to produce samples from a range of domains that they use to help fine-tune their models, which they show helps improve OOD robustness.

Reviewers generally praise the benchmark and point to the corresponding analysis and potential usefulness as a significant motivation for accepting the paper.  These contributions also are generally reliant on the benchmark- if there are significant concerns about the benefit of the benchmark, then it cannot be effectively used to evaluate the proposed DG method.  In this, the paper has significant outstanding issues as it ignores key related work that were both published online on or before Dec 2024 (and, thus, would be considered prior work):

[A] Is Large-Scale Pretraining the Secret to Good Domain Generalization? ICLR 2025

[B] Rethinking the Evaluation Protocol of Domain Generalization. CVPR 2024

At a high level, both papers make the same core observation as this paper: methods that use models trained on large web-scale data like CLIP may overestimate DG performance due to data leakage.  [A] uses this observation to separate samples within a dataset to those learned during pretraining vs. those that don't (i.e., it accounts for a setting where a sample was seen, but not effectively learned), and then analyzes the effect on DG methods. [B] suggests to evaluate methods when they are not initialized on large scale transformer's pretrained weights.  The proposed paper takes a sort of middle ground, where they use the large model (like [A]), but then unlearn features (making it more similar to [B]).  As such, this paper is clearly different than either [A] or [B].  However, what is missing is whether these differences are meaningful.  Can we learn something about evaluating DG methods that we couldn't with [A] or [B]? A clear way of doing that is to evaluate methods on all 3 protocols and analyze any differences in method ranking.  Without this type of analysis, there remains a possibility that this paper simply echoes the same points made by [A,B].

Additionally, [A] and [B] both represent ways methods may be used in practice, where [A] represents applications that use large models and [B] represents applications that are trained from scratch or in substantially different domains (e.g., medical or biological applications).  However, there are few, if any applications that would aim to purposefully unlearn features that might be useful in a downstream domain.  As such, the proposed evaluation setting is more unnatural than the protocols used by [A] and [B].  Thus, even if the authors found their evaluation protocol produced different method rankings than [A,B], the authors would then have to demonstrate that these benefits arise in practice rather than being an artifact of their unnatural evaluation protocol.

Due to these concerns, the ACs find that this paper is not yet ready for publication until its contributions are clarified.  The authors are encouraged to consider these points and others made by the reviewers carefully when revising their paper before resubmitting to another venue.